# Citizens’ Juries: When Older Adults Deliberate on the Benefits and Risks of Smart Health and Smart Homes

**DOI:** 10.3390/healthcare7020054

**Published:** 2019-04-01

**Authors:** Neil H. Chadborn, Krista Blair, Helen Creswick, Nancy Hughes, Liz Dowthwaite, Oluwafunmilade Adenekan, Elvira Pérez Vallejos

**Affiliations:** 1Division of Rehabilitation, Ageing and Wellbeing, School of Medicine, The University of Nottingham, Nottingham NG7 2TU, UK; 2KMB Consulting, Nottingham NG4 3LH, UK; krista@kristablair.co.uk; 3Horizon Digital Research Institute, The University of Nottingham, Nottingham NG7 2TU, UK; Helen.Creswick3@nottingham.ac.uk (H.C.); ezzed@nottingham.ac.uk (L.D.); mszep@exmail.nottingham.ac.uk (E.P.V.); 4Human Factors Research Group, The University of Nottingham, Nottingham NG7 2RD, UK; ezznah@exmail.nottingham.ac.uk; 5Division of Psychiatry and Applied Psychology, School of Medicine, The University of Nottingham, Nottingham NG7 2TU, UK; msxoa15@nottingham.ac.uk; 6NIHR BRC Mental Health, The University of Nottingham, Nottingham NG7 2TU, UK

**Keywords:** smart health, older people, co-design, digital life-world, smart cities

## Abstract

*Background*: Technology-enabled healthcare or smart health has provided a wealth of products and services to enable older people to monitor and manage their own health conditions at home, thereby maintaining independence, whilst also reducing healthcare costs. However, despite the growing ubiquity of smart health, innovations are often technically driven, and the older user does not often have input into design. The purpose of the current study was to facilitate a debate about the positive and negative perceptions and attitudes towards digital health technologies. *Methods*: We conducted citizens’ juries to enable a deliberative inquiry into the benefits and risks of smart health technologies and systems. Transcriptions of group discussions were interpreted from a perspective of life-worlds versus systems-worlds. *Results*: Twenty-three participants of diverse demographics contributed to the debate. Views of older people were felt to be frequently ignored by organisations implementing systems and technologies. Participants demonstrated diverse levels of digital literacy and a range of concerns about misuse of technology. *Conclusion:* Our interpretation contrasted the life-world of experiences, hopes, and fears with the systems-world of surveillance, efficiencies, and risks. This interpretation offers new perspectives on involving older people in co-design and governance of smart health and smart homes.

## 1. Background

Smart cities is a public-policy term for the move towards cities with an increasingly digital infrastructure that enables the real-time monitoring and management of key services in response to changing contexts, typically within transport and traffic management, energy, water, waste, and healthcare. The latter is becoming an increasingly significant area, with “smart health” being a newly coined term to describe the emerging health paradigm enabled by such an infrastructure. According to Solanas et al [1], “Smart health (s-health) is the provision of health services by using the context-aware network and sensing infrastructure of smart cities.” Indeed, with an increasing proportion of the population being over 65 years of age [2], and with continuing constraints on resources, assumptions are made that digital technology will be the solution to improve the lives of older people whilst also reducing health and care costs [2] (p.9). Indeed, being able to deliver “smart”, efficient, personalised health solutions data is key to communicating with users to enableing older people (and their carers and associated health professionals) to monitor and manage their own healthcare and ultimately “age in place” [3].

Despite considerable investment in smart cities, there continues to be low public awareness of the concept. This may be due in part to an overriding emphasis on technology as opposed to engaging with citizens or users; although this focus is starting to shift, as “while citizens tend to be the implied beneficiaries of smart city projects, they are rarely consulted” [4]. Indeed, in an The Institute of Engineering and Technology report [5] in 2016, less than one in five of the general population (18%) were aware of the term “smart city” and only 6% of older people (over 65 years) were aware of the term. The latter, in particular, signals a real challenge when considering the development of healthcare solutions for older people within a smart cities context. It is, therefore, crucial to understand the potential for the involvement of this key stakeholder group, i.e., older people.

It should be said, however, that whilst “smart health” is a relatively new concept, espousing all things digital, data-driven, and connected, there exists a strong body of research relating to more traditional technology-enabled healthcare and assistive technologies (telecare, telehealth, and telemedicine) [6,7] and a wealth of systematic reviews [8]. Nevertheless, despite “people” (older adults, care-givers, healthcare professionals) being the primary focus of such research, there still exists a general lack of understanding of the real needs of such stakeholders, compounded by a further lack of awareness of underlying attitudes, perceptions, and potential barriers to acceptance and use. Indeed, much technology-enabled healthcare research continues to focus on the technical and clinical aspects as opposed to the more subjective conditions of use [9]. There is clearly a need to involve older people/citizens fully in the development of any technology-enabled or smart healthcare initiatives, and ideally at the earlier stages of policy and service development, rather than positioning them as the testers or consumers of technology in pilot or trial settings is crucial [10]. Despite work to engage patients and the public in strategic decision making about health services, there remains a lack of consensus about how such initiatives should operate and which patients should be involved [11].

Research on stakeholder views in the field of telehealth also suggests that there may be a considerable divergence of goals between older people and other stakeholders. In a discourse analysis of 68 publications and 10 knowledge-sharing events on telehealth and telecare, Reference [7] identified four separate competing discourses that tended to “talk past one another”—that is to say, that operated with different assumptions, values, and goals, with little cross-fertilisation. Significantly, they found that these separate discourses tended to map onto different stakeholders, as follows. The modernist discourse was employed by policymakers, the technology industry, and biomedical and health informatics researchers, and it conceptualises technology as the driver and older people as passive consumers. The humanist discourse of older people as active subjects was a separate, more marginalised discourse. Similarly, Peek et al. [12] investigated the aims of different stakeholder groups involved in technology for ageing in place. Whilst stakeholders may agree on aims, the different perspectives held could be problematic in choice and implementation of technology.

The divergence between the views and experience of older people who are being asked to use technology and younger adults who are more likely to be designing and making decisions about implementation of digital technology has been labelled as the “digital divide”. While recognising potential generational inequalities, there is a risk that use of such language and terms such as “digital immigrant” may not be supported by evidence and risk exacerbating stereotypes and stigma [13]. We have interpreted these challenges by drawing on the theory of Communicative Action developed by Habermas [14]. Experience of our personal daily lives, for example our desire for privacy, are part of our life-world, whereas the bureaucratic system of local government and local services tend to form a network of the systems-world. Habermas described the concern of the systems-world encroaching and controlling the life-world, sometimes as a result of corporate interests; this he named colonisation of the life-world. Digital interactions and communication have the potential to form new modes of communication; thus, they have the potential to extend our life-worlds. However, systems are necessarily developed and owned by corporations (private or public); therefore, digital systems are fundamentally systems-world [15]. Taking this perspective, we can consider the negotiation or exchange which may occur, often implicitly, between the individual and the system, in terms of whether digital systems serve the purpose of the life-world or systems-world. 

Partly to counter some of these concerns, co-production or co-design has been advocated as a way to enable end users to have a significant voice and to enable technologies and systems to be designed in a way that is “user-friendly” and accountable to populations (especially local communities). The concept of co-production can be applied to citizen involvement at different stages of the planning, implementation, and review of health and social care solutions [16]. Here, we explore the potential for smart health to be co-produced with older citizens in the UK. 

## 2. Methods

We held initial engagement sessions to co-design the topics and develop personas [17,18] for the citizens’ juries. We then held two citizens’ juries in Nottingham, UK. People who had attended the initial engagement sessions were invited to the second citizens’ jury (B), and therefore, we expected their views and opinions to have developed from the initial meeting. Whereas, for the first citizens’ jury (A), we invited people who were new to the project, and thus, we expected their views may be novel or they may have less awareness of the topics. 

### 2.1. Participants

We engaged with many different stakeholders and networks to recruit participants for the project, including; Vulnerable Adults Provider Network (Nottingham Community and Voluntary Service), Age-Friendly Nottingham Steering Group, Nottinghamshire County Council, Nottingham City Council, Self Help UK and Healthwatch Nottingham. We especially contacted organisations who could help us to reach more vulnerable older residents such as those from Black Asian and minority ethnic communities, and those with disabilities or mental health needs.

For the initial engagement meetings, we also invited staff or volunteers of organisations which engaged with older people. These stakeholders did not participate in jury sessions. In total, 34 people attended these two preparatory meetings. In total three personas were developed but only one was used to prompt discussions within the citizens’ juries.

All participants of the citizens’ juries filled out a consent form, demographics questionnaire and a survey designed to assess attitudinal change before and after each of the citizen’s juries. In total, 23 participants took part in the citizen’s juries: 9 attended Jury A (participants were new to the project) and 14 attended Jury B (participants had previously attended the initial co-design workshop of the project). The age range for both juries was 60–70. Gender was roughly even in both juries, with 4 females in Jury A (44%), and 9 females in Jury B (64%)

### 2.2. Materials and Procedure

The citizens’ jury methodology is described in detail in several studies [19,20,21]. Both jury sessions took the same format over approximately 4 hours including lunch and refreshment breaks. Each session was audio-recorded for later transcription. Participants were first asked to complete a pre-session survey consisting of 9 brief multiple-choice questions which aimed to gauge the level of knowledge participants had and their existing opinions about issues of relevance. These included questions such as “How often do you use technology such as the following: mobile phone, motion sensors or alert systems?” and the possible answers: “Several times a day”; “Sometimes”; “Rarely” or “Never”; or questions such as “Who should design health technology applications for well-being?” and possible answers: “Technology developers”; “Technology consumers”; “Local government”; “All of all the above”; and “Other, please write a few words”, (see Appendix A Jury Post-session Survey for details).

Participants were then presented with a series of dilemmas and encouraged to discuss the issues that each dilemma raised (see two examples below and a summary of topics discussed is shown in Table 1). The topics for the dilemmas were developed in the preparatory meetings. Furthermore, participants were asked for their recommendations on how to address the dilemma or problem presented to them. The juries were all moderated by an experienced facilitator, an adult previously unknown to the participants and who was not presented as an authority figure. The facilitator made sure all participants had the chance to be heard, with all experiences, viewpoints, and recommendations seen as valid and respected by all members of the jury. The sessions were guided in a way that was not leading or instructive so as not to prescribe opinions. Discussions took the form of a deliberation after each dilemma was presented, around two tables of 4 to 7 participants. This allowed participants to share opinions with the emphasis being that there were no right or wrong answers.

Examples of the dilemmas include: 

Safety monitoring versus concerns of loss of independence: Assistive technology and monitoring in the home may benefit people by offering support and to reassure people of safety. However, some people may feel that monitoring implies “keeping tabs” on them and that this may reduce privacy and independence.

Data-sharing and privacy: If someone’s medical information was shared with their social worker then this may avoid duplication of the same questions. On the other hand, there was a concern for privacy; will the individual know and have control over who has access to personal data?

These dilemmas were presented to be discursive rather than prescriptive, to prompt responses and recommendations, and a persona (see Figure 1) was also created as a way to tell a story about how an individual may be affected by digital technologies and how this may affect their health.

This survey (see Appendix A) consisted of 3 brief multiple-choice questions designed to measure attitudinal change, followed by a series of 15 statements designed to measure opinion on the issues raised; 10 statements were scored on a Likert scale from 1 (agree very little) to 10 (agree very much), and 5 were scored on a Likert scale from 1 (applies to me very little) to 10 (applies to me very much). Statements covered similar issues to those from the pre-survey including benefits/risks of health technology for society and perceptions on influencing decision making. 

## 3. Results

### 3.1. Participants’ Demographics (Table 2)

### 3.2. Opinion Survey: Pre-Jury and Post-Jury

This section compares responses from the pre- and post-surveys between the two groups. We were interested in whether participation within the jury led to changes in attitudes, and therefore, we invited people who were new to the project to one group, Jury A, whereas people who had attended the initial engagement meeting, and therefore had experience within the project were invited to Jury B. However, none of the survey differences between juries were significant when applying non-parametric statistic *χ^2^*, thus, prior involvement in the project did not appear to significantly change attitudes. 

#### 3.2.1. Pre-Jury Survey

The pre-session survey revealed that at least two-thirds of the respondents in both juries use technology; the majority use technology several times a day. Additionally, a majority of people in both juries felt it was at least quite important for older people to use new technologies (93.3% of the group who had experience with the project, 66.6% of the group who were new to the project). 

Most respondents in group A, who were new to the project, (85.7%) said that “Smart City Nottingham” made them feel interested about future opportunities. Whilst, in the group who had experience with the project (B), a large proportion of the respondents were split between being interested (44%) and concerned about technology (44%). In regard to the influence smart cities have over the future of healthcare of older people, responses in both group sessions were varied. A large number of the new group (A) did not know how much influence smart cities had (44.4%). Whereas in the group with experience with the project (B), the responses were mixed. This indicates that there were a range of perspectives within both groups. The range of views expressed addresses any concerns that the project may have recruited a self-selecting group; for example, people who were very critical or cynical of digital innovation. 

People of different ethnicities have been described as experiencing a digital divide in a similar way to older people [22]. We have involved participants of different ethnicities and religions, as shown in Table 2, indicating that we have a mixed group of participants; however, we did not aim to analyse these intersectionalities.

A majority of both juries believed that they should have an influence in the designing of assistive technologies (77.8% and 54.5% in the new group and the group with prior experience, respectively). When asked who should design and implement health applications, a majority of respondents on both juries said that this should be a mix of technology consumers and local governments. In regard to whether the respondents thought about the ethical consequences of health technologies, at least two-thirds of both juries revealed that it is something they thought about a least a little bit. 

#### 3.2.2. Post-Jury Survey

Participants were asked to complete a survey immediately after the jury session in order to assess whether topics raised within the discussion had prompted concerns or changes in views. After the session, when asked who should be accountable if smart technologies go wrong, a majority of the group new to the project (A) answered “Other services” (55.6%) with smaller responses opting for the “Manufacturer” and the “Health Services” (Figure 2). Whilst the greatest response of the group who had experience with the project was tied between “Other services” (38.5%) and “Manufacturer” (38.5%). 

When asked if the participants had learnt anything new about assistive technologies, at least two-thirds of both juries said they had learnt at least “A little” (84.6% and 66.6%, in the groups with prior experience and new to the project, respectively).

In regard to whether the participants had come up with new ideas about how to increase accessibility of smart cities for older people, a majority of respondents in both juries reported that new ideas emerged during the sessions (69.2% and 66.7% in the groups with prior experience and new to the project, respectively), whilst around a third in both juries reported no new ideas had emerged during the sessions.

### 3.3. Analysis of Discussion During the Citizen’s Jury Sessions 

The deliberations that took place at the two citizen’s jury sessions were audio-recorded and analysed through exploring two perspectives. Personal experiences as well as hopes and fears about how technology may affect individuals was interpreted as reflecting the life-world. Participants discussed the potential efficiencies or improvements that the digital system could achieve; they also expressed concerns about surveillance of citizens and other risks, and these were interpreted as reflecting the systems-world. These two perspectives enabled a more nuanced interpretation, rather than a polarised interpretation of positive or negative outcomes (for the individual). Initial themes emerged from the groups of the open-space engagement session. The discussions within citizens’ juries then added weight and resonance to these (see Table 3).

### 3.4. Concept of Smart Health

There was much discussion about the meaning of the term “Smart Health”. Our assumption was that the term relates to digital technologies that may improve or affect health and healthcare, and much of the discussion resonated with that concept. Different interpretations were that SMART was an acronym for something or that smart meant healthy living, or equivalent to good health literacy.
“…it’s what you eat. Now then isn’t that an education process where we’re talking about being smart with our health? It’s nothing to do in essence we’ve got a gizmo on the table; it’s whether or not we’ve got the capability to understand what in fact smart health is.”(Group A, male respondent)

Whilst this quote initially appears to be discussing a different concept; it highlights a need to understand health literacy as well as digital devices. Having considered this range of concepts of the neologism “Smart Health”, we will focus our interpretation on the meaning that many participants touched upon. This was very clearly described in the following quote from one participant:
“…about using devices like your mobile phone, your computer, an iPad-kind-of-thing, anything digital like that. And then using like little programmes that you might call apps with some computers to help you manage your health long term of your life. So that if you’ve got a health condition like diabetes or something, you can manage it yourself and take control and be independent, but I would only say that as an abstract concept, not as a living position.”(Group B, female respondent)

For the main part of the discussion, we interpreted views about a number of topics, and we have attempted to contrast two perspectives that were voiced by participants; views about personal experiences, or life-world, and views about the system or citizens as a whole.

### 3.5. Theme 1: Control and Privacy versus Mis-Trust in Purpose of Data Use

Discussion about errors and fraud were voiced as a way to demonstrate concerns about control and privacy. One participant describes the GP software system being offline, possibly due to an error, and this preventing transfer of case notes. This may be frustrating at a personal level, due to inconvenience, but it may prompt general concerns about risks of data, due to error or fraud.
“At the moment the software at my GP place is—to use a technical term—buggered up, because I’ve got some other thing and they won’t transfer electronically.(Group A, male respondent)

One participant had concerns about the Council using or sharing data in ways that were not in the interests of the individual. Concerns were raised about whether data was being collected in order to develop a marketable database of personal data. This indicates an awareness of the high value of personal data and also a lack of trust in the purpose of the system collecting this data.
“…I have a comment on the technology of this. That is, I think our approach is entirely wrong. The technology is being introduced so as to accumulate a large databank which is sellable; it’s not got anything to do with our health.”(Group A, male respondent)

There was discussion of governance and suggestions of additional regulation to reassure individuals. There was also an acknowledgement that there may be a diversity of views from individuals about the level of concern about sharing data.
“I have no problem personally with sharing my data, but I do understand other people do. And it’s a matter of choice. For me the solution to this would be actually regulation. So, if people abused access to your data and information that there were penalties that they would pay.”(Group A, female respondent)

Concern about private multinational companies collecting medical data.
“…Google are now wanting to set up a website to do with smart health. They want access to your medical records, and I’m against that, some people who agree with it, that’s entirely up to them, but with me my information will stop with the people who I want to have my information.” (Group A, male respondent)

In this section, experience of digital technology in the personal life-world may be a feeling of invasion of privacy of data, especially if an individual’s data is being used or shared in a way that was not clear or transparent. Furthermore, digital technology may enable an individual to have a greater sense of control of their GP appointment, for example, but when an error occurs, this might spark concerns about a lack of control of their personal medical data. On the other hand, the weaknesses of the systems are revealed when a computer (ICT) problem occurs, which leads to loss of control. Where the system shares data, there may be concerns as to the purpose. A concern about the systems-world is that it gathers data, almost as an inherent characteristic. Beliefs about motivation for collecting data were because large datasets are seen to be valuable or because data could be used to control or surveil the individual. 

Within this theme, the life-world perspective may be described as the convenience of using online systems, for example booking appointments or sharing data with different professionals. Whereas the systems-world perspective highlights a concern that personal data is being amassed, and this may be associated with risks of accidental breach of confidentiality, or purposeful selling of data. There was also a concern that data could be shared with a motivation of controlling aspects of people’s lives (maybe welfare benefits) or services. Responses to these concerns were at both the systems-world level (regulation and sanctions) and the life-world level of acknowledging that people opt-out or refuse to share their personal data.

### 3.6. Theme 2: Choice or Self-Efficacy versus Standardisation

Fears were voiced that with an increasing implementation of digital systems, in the future it will not be possible to opt-out or use non-digital processes. This may be interpreted as the efficiencies of standardisation of the systems-world; that bureaucracies aim for a standard process rather than flexibility to individual preferences. Participants suggested that some individuals may not want to use digital technology; which indicates that there is an expectation within modern discourse that everyone will adapt to digital technology (given time and opportunity). The views expressed questions of whether some people may not accept digital technology, and whether their views and rights should be respected. This led to an expression of concern that a group of people may have their rights infringed upon in the future, and that they will be disadvantaged if they do not accept the use of digital technology. One participant used the analogy of online shopping:
“It’s like people who buy things online now and get a better deal. But not everybody wants to do that, and not everybody should be forced to do it. So, it might be … based on individual need and the individual willingness to do it.” (Group A, female respondent)

This description of buying goods online as an analogy to accessing welfare services indicates an acceptance of the discourse, in media and policy, about welfare services being conceptualised as commodities to be bought by, or given to, individuals, rather than as public goods to which citizens have a right to access. This is exemplified by the phrasing of this quote: “…manage for yourself; your health, your wellbeing over a long time” (Group B, female respondent).

The systems-world perspective is often about standardisation and efficiency of processes and services. Thus, there was a view that, in the future, older people would not have a choice, but would have to use digital technology to access health and care services.
“I think there is a certain section of society upon which it will be imposed. They won’t have any choice, mainly for cost reasons. Services can only operate if we have a system working and everyone is included in it...the point will come when they cannot be cared for adequately without this system, without wearing something on their arm. And that will come with our 87-year-old [persona]. If she hasn’t taken her chance to learn basic technology when younger, when she is older and very dependent, she’s so confused she doesn’t know how to use it, and she hasn’t a position to say no I resist any longer. It will be forced on her; she will have to accept it. So, it will be unfair, it will be undemocratic, but that is the way it is likely to go.” (Group B, male respondent)

This respondent makes a clear link between the systems approach of standardising care processes and the risk that this may mean that some individuals will have to accept technology with which they do not feel comfortable. At a personal, life-world level, this indicates a constraint in choice of care or treatment, while at a systems-level this becomes about democratic choice in investment in services and technologies. 

### 3.7. Theme 3: Data Sharing Enables Continuity of Care versus Cross-Checking between Agencies

One participant described data-sharing in a positive way; this participant is describing telehealth.
“…if you’re wearing or having some device, then the information you provide or is provided by you, or your piece of equipment, then goes back to a centre. So, it goes to your health worker, whether it’s your GP, the hospital, district nurse or whatever they call them today, and that saves time, energy, money.” (Group A, female respondent)

The participant implies that through sharing data between all members of the healthcare team, it will improve efficiency of communication, and hence improvement of continuity of care.

However, another participant had a very cynical view of how organisations could use personal data.
“…if you ever have a problem with [organisations] like I do, they can access your whole life near enough at the click of a mouse button. And I don’t want them to have that.” (Group A, male respondent)

This participant mentioned that he had previously had conflicts with the Council, so this may have shaped his mistrust in the digital information. This demonstrates how views about digital or smart technology are overlaid on previous relationships with institutions; these might be new technologies, but they are embedded in existing bureaucracies and systems.

These two respondents demonstrate how this interpretation may open new discourses about data use and trust in data-sharing. Whereas the first quote is about personal care and improving continuity, that is where the individual may gain benefits from opting into the system. The second quote shows how the individual is thinking about how the system works at a bureaucratic level, and what the implications might be for control of personal data. Development within smart cities should acknowledge these two discourses in order to improve governance and processes as well as communications about these with stakeholders and public.

### 3.8. Theme 4: Systems-World Reach into Personal Devices; Convenient Reminders or Over-Reach? 


“…the appointments, notifications on the phone. Which I think is great, it’s a good idea.”(Group A, female respondent)


This participant is describing the healthcare system’s use of efficient scheduling and digital communication to reach into the domain of personal communication, the mobile phone. The participant welcomes this, presumably from a perspective of convenience and preventing forgetting the appointment. However, this may be an area of tension, where other individuals may feel that reminder notifications on their mobile phone may invade their personal space and life-world. Another participant had had phone and skype consultations with the doctor and this participant had a similar view; that this was convenient and saved the doctor’s time.
“I very often don’t need to go down to the doctor. I’ve had one phone appointment with the doctor, but I would quite like a Skype for the next time appointment; to save me going down sometimes and to save them time.”(Group A, female respondent)

Again, receiving a phone call from the doctor at home and conducting a medical consultation over the phone could be perceived as the systems-world accessing the personal space of home, and carries the risk of communications being unsecure. People may become concerned that organisations or systems can reach into their personal space to communicate or monitor their activity.
“…Even though I’ve got a laptop, I treated myself to a [Smart TV]…it frightens me to death. I’ve got this thing that somebody’s watching me.”(Group B, female respondent)

For individuals with limited cognition or communication, it may be difficult to understand their view on health monitoring and use of data; and yet this may be a situation where monitoring an individual’s health status is a priority. One participant described the importance of understanding the individual’s wishes before cognitive decline.
“I know my husband and I have talked about people having power of attorney at various time about care, about finances. People have got to make those kinds of decisions before they…[deteriorate].”(Group A, female)

### 3.9. Theme 5: Ownership versus Collecting Population-Level Data 

Participants from one session mentioned ownership of health records, comparing the situation in Britain with France. Her experience in France was that individuals have ownership of their records and take them to the doctor, whereas Britain was perceived to be behind the times in not enabling people to own their records.
“…why Britain is one of the few countries in Europe that people don’t keep their own records. I mean I know that when I’m in France if someone goes to the doctor, they take their records with them. And I don’t see why I’m not grown up enough to know what’s wrong with me…in Britain, it’s always been the doctor’s always the way; that knows the answer, and you’re there listening to the great God doctor.” (Group A, female respondent)

This participant is indicating that the lack of access and ownership to personal health data indicates an entrenched paternalistic relationship between healthcare professionals and patients. This is a description of the systems-world, and a frustration that the personal health information cannot be owned and co-located within the life-world of the individual.
“…I think it is important that the individual is in charge of it.” (Group A, female respondent)

Ownership of data could lead to individuals checking the validity of data and correcting errors. Another respondent indicates that they would be willing to share personal data, as long as an appropriate regulatory framework was in place, with appropriate sanctions.
“…I have no problem, personally, with sharing my data, but I do understand other people do. And it’s a matter of choice. For me the solution to this would be, actually, regulation. So, if people abused access to your data and information, that there were penalties that they would pay.” (Group A, male respondent)

An exchange between two participants highlighted the difference between personal data for care of the individual compared to the same data being aggregated and used for population intelligence. The first participant starts by introducing the idea that information is provided by the individual, phrasing which may indicate a sense of ownership. This information then “goes back” to a centre which coordinates professional activity; this phrasing suggests a spatial distance between the personal and professional (systems) worlds. The outcome of these processes is that “your GP…district nurse” is notified of the issue and can respond in an efficient and timely manner, indicating a personal and convenient response. These savings may refer to the system, and the mention of money suggests efficiency for the system rather than savings for the patient (as there are no out-of-pocket fees for health professionals’ time in the health service in England).
“…if you’re wearing or having some device, then the information you provide or is provided by you, or your piece of equipment, then goes back to a centre. So, it goes to your, so your health worker, whether it’s your GP, the hospital, district nurse or whatever they call them today. And that saves time, energy, money.” (Group A)

In responding to this participant, another participant takes the “indirect” perspective of the systems-world. He argues that although there has to be potential to benefit the individual patient, there also has to be a benefit for the health system; this phrasing—“has to benefit the health service”—suggests a “business case” type of argument. Personal data collected by various devices is interpreted by analysts to yield population data in order to improve decision-making for future health service planning. This latter perspective is an objective argument which also has potential to benefit the individual in the long term, and is a strong contrast to the personal benefits of arranging multi-disciplinary care in a timely way to meet the needs of an individual (person-centred care).
“It has to be for the benefit of the patient. I fully accept that. But, also, there is an indirect benefit to the patient in that it has to benefit the health service itself. The collection of data about the community—and that will ultimately help you. It may not give you an immediate assistance, but down the line, people who are able to interpret it will know more about the population and be able to make more intelligent decisions about healthcare.” (Group A, male respondent)

Taking a systems-world perspective, the participant argues that aggregated data can inform health planning. This is a complex argument and indicates a high level of knowledge and consideration by this particular participant.

### 3.10. Theme 6: Co-Design for Older People

Different perspectives may be characterised as “why do we have to use digital technology to access services that we had for years”. This contrasts with the systems-world assumption that older people should use technology in the same way as younger people (already) do.
“Now does in fact Gladys [persona] want somebody to call in to her who can remind her how to in fact access a part of a computer programme? I forget, and I’d spend more time trying to remember how to do it, purely and simply because I only need to do that particular problem on an infrequent period of time. So, I get frustrated.” (Group A, male respondent)

With this perspective in mind, participants were keen that technology developers should involve older people into the design of products and systems.
“But the technology companies have to employ people like Gladys [persona] and say right, we’ve got this thing, does it work for you? And I’m not sure the extent to which they use people like that when they’re designing their products.” (Group A)

Thus, at the systems-level, data might identify that a proportion of people are not accessing technologies or services delivered in a technological context; however, we need to understand how individuals interact and find meaning in digital technologies, in order to improve design to be accommodating of all older people.

## 4. Discussion

### 4.1. Key Findings

This paper compares the opinions and attitudes about smart cities and the impact on health and well-being. We held two citizens’ juries, where the difference between the two juries was that one group had previously been involved in the co-design of the content of the session (B), whereas the other group were new to the project (A). The results revealed that there were no differences between the juries in existing levels of knowledge, opinions, and in attitudinal change. The pre-session survey was implemented to gauge the existing level of knowledge and opinions. Whilst the post-session survey was implemented to measure attitude change and measure opinions on the issues discussed.

The survey completed before and after the jury session can be linked to the topics discussed at the juries. Discussions revealed participants’ deliberations about the benefits and risks of smart health technologies and system. During the pre-session survey, 44.4% (Jury A) and 14.3% (Jury B) of participants expressed concerns towards technology. This result highlighted the differing welcoming attitudes to smart health. Whilst discussing attitudes, participants voiced scepticism and resistance towards smart health technologies. Concerned participants expressed a preference for face-to-face support. In the post-session statements, a majority of participants did not agree it was a good idea to replace humans with technology. However, the participants did express that technologies can help reach those who live alone and aid in social interactions, mentioning benefits to health problems in older adults such as dementia. This was reflected in the over half of the participants agreeing that smart city initiatives can help reach more people. 

Although a majority of participants suggested in the pre-session survey that they often use technology, in the discussion, participants made recommendations of training in technology for older adults. They also mentioned issues of the digital divide, which was expressed in rating in post-session statements. Despite identifying a digital divide, pre-session results suggest that participants do believe it is important that older adults use new technologies. Furthermore, responses to the post-session questionnaire which suggests that individuals will try and use health technologies more often, although responses were mixed. This should be an incentive on the potential of greater use of health technologies, provided technologies are accessible, simple, and affordable for the target population. 

Participants recognise the importance of sharing information through health technologies and how it can potentially benefit their navigation in healthcare (such as making appointments). A majority agreed in the post-session statements that the benefits of health technologies exceed the risks. The group did, however, raise issues relating to the regulation of data sharing and their part in controlling the information. Generally, in the pre-session survey, when asked about ethical consequences, a majority of participants in both groups had some concern. This trend continued in the responses in post-session statements related to ethical consequences, where even after discussion, a majority disagreed with having minimal concern. 

Qualitative analysis used a Habermasian approach of exploring perspectives on life-world and systems-world. The advantage of this approach is that the personal experience can be investigated and separated, to a degree, from the qualities of the emergent system. This is particularly important with integrated systems and data; it may not be the individual piece of technology which has a positive or negative outcome, but rather the technology within multiple interrelated systems (digital and process, i.e., bureaucratic systems). We applied this approach to explore six themes which were prompted or emerged during the citizens’ jury sessions. 

While raising some scepticism and concern, participants generally want to be more engaged in the design and implementation of health technologies. The participants stressed the importance of testing technologies on older adults, echoing ideas that technologies need to be simple and accessible. This collaborative approach reduced concerns of being forced to engage with technologies that are not wanted and allowing older adults to regain control. 

### 4.2. Internal Validity—Strengths and Weaknesses

A range of views were expressed from male and female respondents and across the group which had previous involvement in the project and the group which was new to the project. No specific patterns were detected across these groups. Furthermore, we did not detect a self-selection issue.

These were small groups (*n* = 9 and *n* = 12) from one city in England. The views were likely to be influenced and contingent on the public discourse within the city. However, this approach was important to recognise for a city-based initiative such as “Smart City Nottingham” because processes and public communications should be adapted to local contexts. 

We took a very broad approach to digital technology, rather than focusing on a particular platform or device. The weakness of this approach was that various comments may not relate, and there may be a lack of depth of discussion. However, the advantage was that the analysis gained a “bigger picture” interpretation of concerns which may be important to understand at an overarching level.

### 4.3. External Validity—How Does It Compare to the Literature

Our broad approach relates to many different disciplines, from healthcare to data-systems design. This approach is consistent with “lifeworld-led healthcare” and the previous body of work on patient-centred care [23,24]. We have built on a Habermasian analysis of the medical encounter where the intermediary between doctor and patient, a language interpreter, implicitly negotiates between life-world and systems-world [25]. Whereas, in our study, digital technologies and systems act as intermediaries between citizens and health professionals and the city bureaucracy. This approach has enabled a detailed interpretation of complex interrelationships which are often conceptualised as a “wicked problem” of the “digital divide” [26]. 

### 4.4. Future Work

Whilst some discourses perpetuate the view that older people respond in a passive way to innovative technology, our study has found a desire of individuals to be consulted and participate in the co-design of smart systems. There is a growing awareness of potential inequalities that may emerge as older people find it difficult to access services due to technological barriers. From a human rights perspective, older people have a right to be involved in the design and implementation of technologies and systems where they are the main beneficiary. Further work is needed to explore the two elements of health literacy and digital literacy and how these interact at a personal level and at a city-wide level.

## 5. Conclusions and Recommendations

Our study took a co-design approach in developing citizens’ jury sessions to explore the views of how a smart city may affect people’s health and well-being. Using a persona to discuss several dilemmas enabled exploration and deliberation on a number of common themes of data control, privacy, and convenience of technology. Surveys before and after the jury sessions captured the range of perspectives within the group and could counter any claims that these groups of participants represented any particular interest. Participants expressed concerns about the risks of data sharing and use of data; however, the convenience of booking appointments or accessing online healthcare records was valued. Participants were aware of the benefits of digital systems to the health and care sector, especially for efficiency and collection of data. Our interpretation of life-world and systems-world perspectives enabled a nuanced understanding of these tensions or trade-offs within the implementation and experience of a smart city for older people. 

We recommend further research in the following topics that were found to resonate with participants: data-sharing and trust in use of data; personalisation or standardisation; and surveillance in the home. Many of these topics relate to trust between citizens and the organisations involved in the system (especially health and social care providers). Co-production may facilitate trusting relationships, and citizens’ juries are one method to achieve this with a rights-based deliberative consultation. Further research is required to explore how statutory, private, and third-sector organisations can best respond and incorporate these views in strategy and implementation.

## Figures and Tables

**Figure 1 healthcare-07-00054-f001:**
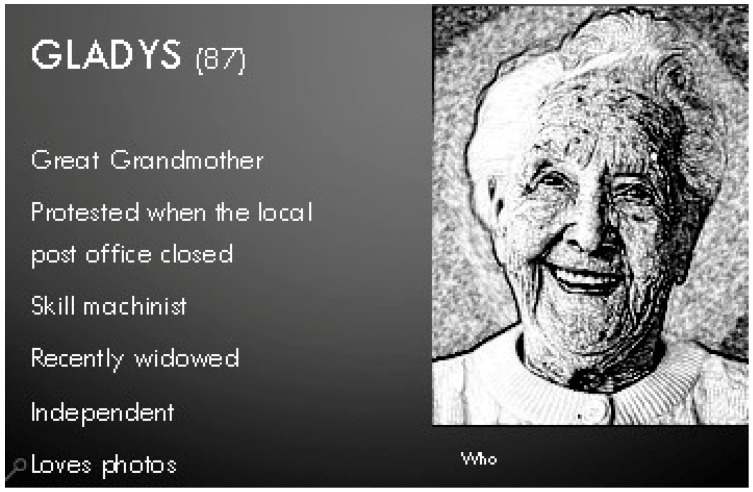
Persona created as a way to tell a story about how an individual may be affected by digital technologies and how this may affect their health.

**Figure 2 healthcare-07-00054-f002:**
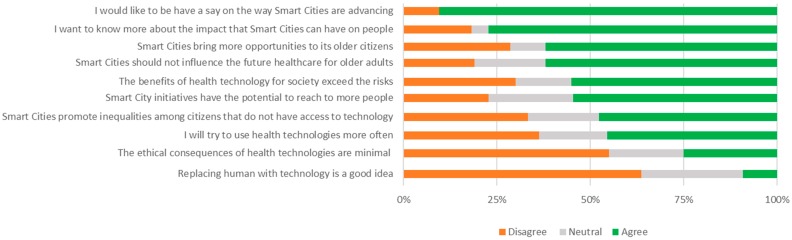
Post-session responses of all participants to the statements relating to issues raised in the juries. Differences between the groups were not significant for any of the statements.

**Table 1 healthcare-07-00054-t001:** Table of topics and dilemmas discussed within the citizens’ juries.

Topic	Issue or Dilemma
Smart health concept	Does the term smart health resonate or carry meaning?
Sharing of personal (medical) data	Ownership of data and continuity of care or risk of misuse?
Online systems to access health or social care	Convenient or barrier for some people?
Digital technology in the home	Reassurance for family member or invasion of privacy?
Barriers to access	Cost barrier of digital devices, lack of broadband internet connection?

**Table 2 healthcare-07-00054-t002:** Self-reported characteristics and beliefs of participants.

Total Participants (*n* = 23)		Jury A (*n* = 9)	Jury B (*n* = 14)
Gender	Female	44% (*n* = 4)	64% (*n* = 9)
	Male	56% (*n* = 5)	36% (*n* = 5)
Age	Younger than 60	0	0
	60–70	44% (*n* = 4)	50% (*n* = 7)
	70–80	44% (*n* = 4)	36% (*n* = 5)
	Older than 80	2% (*n* = 1)	14% (*n* = 2)
Religion	No religion	56% (*n* = 5)	29% (*n* = 4)
	Christian	33% (*n* = 3)	57% (*n* = 8)
	Unitarian	11% (*n* = 1)	0
	Wiccan	0	7% (*n* = 1)
	Prefer not to say	0	7% (*n* = 1)
Activity limitation	Very limited	2% (*n* = 1)	44% (*n* = 4)
	Limited	44% (*n* = 4)	0
	No	33% (*n* = 3)	50% (*n* = 7)
	Prefer not to say	2% (*n* = 1)	21% (*n* = 3)
Health	Good	22% (*n* = 2)	57% (*n* = 8)
	Fair	88% (*n* = 7)	36% (*n* = 5)
	Bad	0	7% (*n* = 1)
Ethnicity	White British	100% (*n* = 9)	72% (*n* = 10)
	White Other	0	7% (*n* = 1)
	Caribbean	0	21% (*n* = 3)

**Table 3 healthcare-07-00054-t003:** Topics which emerged during workshops.

Topic Number	Personal, Life-World	Strategic, Systems-World
1	Control, privacy	Mis-trust about purpose of data collection, lack of control
2	Choice, access to information and personal efficacy	Standardisation, paternalistic
3	Continuity of care is benefit of information sharing	“Using data against you”, e.g., cross-checking between agencies
4	Monitoring for safety	Surveillance and utility of data, reaching into personal domain (e.g., mobile phone)
5	Ownership	Population collective data of public sector data
6	Experience of technology in older life	Lack of adjustments for older people

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
