# Peer review of "Citizens’ Juries: When Older Adults Deliberate on the Benefits and Risks of Smart Health and Smart Homes"

_healthcare, 2019, doi:10.3390/healthcare7020054_

Round 1

Reviewer 1 Report

Overall this is a good paper that is generally very well situated and justified in a current body of knowledge. I suggest a few minor amendments: 

I agree that people are being left out of the technologically -driven discourse on topics such as smart cities and their associated products. I wonder whether a little more insight into why this is the case in the introduction would be enlightening to the reader. Why doesn't the human element get looked at in such research? And secondly, why don't older people get more of a look in do you think?

Could you explicitly say why you ran two citizen's juries with different participants? What was your aim here and why did running two resolve this aim? More detail and justification would again be helpful here to people repeating a similar study and for people reading and interpreting the findings for themselves. 

You describe " Black Asian and Minority Ethnic communities and those with disabilities, or mental health needs" as being more vulnerable. Can you say why you think they are more vulnerable in this setting? Is there evidence to suggest they are? Be explicit again here as to why.

Can you justify why you invited Staff or volunteers of organisations to take part along with older people? Isn't there a risk that more "professional" advocates would take over conversations? Or did you want their wider more broad insights? Again all that is needed is some explanation or justification.

I'd like to see the dilemmas presented - how many were there? - and what subejcts did they debate and why they were chosen - are they based around previous knowledge/literature in the field or something else? 

I like the way the themes are presented and discussed but I found some of the more quantitative data difficult to read - maybe figure 2 could be improved by having them in order of % agreeing.

I love how skeptical the participants were and there is some lovely illustrated quotes to accompany this! 

I am happy with the discussion that brigs out the relevant key points and discusses the limitations well. I am happy with the conclusion too.

Author Response

R1- Q1:

We thank the reviewer for their helpful constructive comments. We have added text to expand on the human-element – engaging with patients and public, as follows:

“Despite work to engage patients and public in strategic decision-making about health services, there remains a lack of consensus about how such initiatives should operate and which patients should be involved (10)"

Continuting Q1 and also responding to Q3

We have also expanded on the digital divide comment, questioning the validity of such terms:

“While recognising potential generational inequalities, there is a risk that use of such language, and terms such as ‘digital immigrant’, may not be supported by evidence and risk exacerbating stereotypes and stigma (12).  “

We have added the following with respect to ethnicity and religion:

“People of different ethnicities have been described as experiencing a digital divide in a similar way to older people (21). We have involved participants of different ethnicities and religions, as shown in table 1, indicating that we have a mixed group of participants, however we did not aim to analyse these intersectionalities.”

R1-Q4

Stakeholders were invited to the initial consultation meeting where the topics were developed, but did not participate in jury sessions, we have added:

“These stakeholders did not participate in jury sessions.”

R1-Q5

We have inserted a table of all topics discussed (Table 1).

We stated in methods: “We held initial engagement sessions to co-design the topics and develop personas” and again in materials and procedure: “The topics for the dilemmas were developed in the preparatory meetings.”

Reviewer 2 Report

despite the small number of participants, the study obtains relevant results. Well organized and presented. 

Author Response

We thank reviewer 2 for positive comment about the manuscript.

Reviewer 3 Report

I consider this study about affect people’s health and wellbeing is very interesting because took a new co-design approach in developing citizens’ juries.

For this reason, authors must take special care to preserve the purpose of the study: To facilitate a debate about the positive and negative perceptions and attitudes towards health digital technologies.

In the manuscript. Thus, in the manuscript, the authors should explain in detail: 

1. ¿How Religion influence the study carried out and the results presented? (Why was it asked and not analyzed?)

2. ¿How Ethnicity influences the study carried out and the results presented? (Why was it asked and not analyzed?)

3. With whom (government, institutions, supplier, staff...)  the debate was held about the positive and negative perceptions and attitudes towards health digital technologies

4. Results and Conclusions: Maybe in a separate analysis more clear:

¿Which are the benefits and risks of smart health technologies and systems?

5. Conclusions: ¿What was interpreted from a perspective of life-worlds versus systems-worlds? 

Explain why: "The authors recommend further research in the various themes that emerged during this jury sessions? What topics?

Author Response

We thank the reviewer for their helpful constructive comments which we have addressed as follows:

R3 – Q1 & 2: We have added the following with respect to ethnicity and religion:

“People of different ethnicities have been described as experiencing a digital divide in a similar way to older people (21). We have involved participants of different ethnicities and religions, as shown in table 1, indicating that we have a mixed group of participants, however we did not aim to analyse these intersectionalities.”

R3 – Q3: Stakeholders were invited to the initial consultation meeting where the topics were developed, but did not participate in jury sessions, we have added:

“These stakeholders did not participate in jury sessions.”

R3- Q4 : We have not aimed to clearly state benefits and risks in this study. We have taken a lifeworld/systemsworld approach to try to explore the complexity, where benefits to system may be risks to individual. Or benefits perceived by one individual may be perceived as risks to another individual.

R3-Q5: To conclusions we have added:

“Participants expressed concerns about risks of data sharing and use of data, however the convenience of booking appointments or accessing online healthcare records was valued. Participants were aware of the benefits of digital systems to the health and care sector; especially for efficiency and collection of data. Our interpretation of lifeworld and systemsworld perspectives enabled a nuanced understanding of these tensions or trade-offs within the implementation and experience of Smart City Nottingham, for older people.

We recommend further research in the following topics that were found to resonate with participants; data-sharing and trust in use of data, personalisation or standardisation, surveillance in the home. Many of these topics relate to trust between citizens and the organisations involved in the system (especially health and social care providers). Co-production may facilitate trusting relationships, and the citizens’ juries is one method to achieve this with a rights-based deliberative consultation.”